# Telehealth equity and access communication skills pilot simulation for practicing clinicians

**Christopher J. Nash**[1]*, **Susan E. Farrell**[2], **Jossie A. Carreras Tartak**[3], **Alexei Wagner**[4], **Lea C. Brandt**[5], **Emily M. Hayden**[6]

1 Department of Emergency Medicine, Duke University Hospital, Durham, North Carolina, United States of America, 2 Harvard Medical School, Boston, Massachusetts, United States of America, 3 Department of Emergency Medicine, Beth Israel Deaconess Medical Center, Boston, Massachusetts, United States of America, 4 Department of Emergency Medicine, Brigham and Women's Hospital, Boston, Massachusetts, United States of America, 5 Center for Health Ethics, University of Missouri School of Medicine, Columbia, Missouri, United States of America, 6 Department of Emergency Medicine, Massachusetts General Hospital, Boston, Massachusetts, United States of America

* chris.nash@duke.edu

**Data Availability Statement:** All relevant data are within the manuscript and its Supporting information files.

## Abstract

### Objectives

This pilot study evaluated a telehealth training simulation program for practicing clinicians, specifically focused on addressing patient issues of equity and access to healthcare via improving telehealth communication.

### Methods

Participants participated in a one-hour simulation experience with two cases. Performance was assessed pre- and post-intervention using a checklist measuring communication domains related to equity and access in telehealth. Participant satisfaction was secondarily measured via survey.

### Results

Results showed measurable gains in clinicians' abilities to effectively incorporate equity and access communication skills. Participants found the session useful and recommended the training experience.

### Conclusions

The findings of this pilot study highlight the potential of simulation-based telehealth training for practicing clinicians, emphasizing clinicians' attention to patients' equitable access to healthcare. Future studies should aim to explore the durability of learning and investigate the generalizability of this training approach to other telehealth competencies and settings.

**Funding:** This research was financially supported by the Association of American Medical Colleges Telehealth Equity Catalyst Grant (https://www.aamc.org/news/telehealth-equity-catalyst-awards), grant number N/A, with authors CJN SEF and EMH as recipients. The funder had no role in study design, data collection and analysis, decision to publish, or preparation of the manuscript. There was no internal funding received for this study. There was no additional external funding received for this study.

**Competing interests:** The authors have declared that no competing interests exist.

## Introduction

As telehealth becomes more integrated into healthcare delivery [1–4], it is increasingly important that healthcare professionals learn the necessary skills to effectively care for patients in the virtual environment. Healthcare delivery via telehealth presents distinct challenges compared to the traditional care environment, and training clinicians for this care modality requires instructional techniques specific to its unique demands [5, 6]. Recent evidence underscores the need for defining telehealth competencies to guide training programs, ensuring that clinicians across allied health professions are equipped to navigate the technological, ethical, and communication challenges inherent in virtual care [7–12]. The Association of American Medical Colleges (AAMC) created core competencies defining specific skills including conducting a virtual physical examination, troubleshooting technological failures, and communicating through a virtual connection [8, 13, 14]. The AAMC telehealth competencies serve as the best guide to-date for designing and implementing curricula for telehealth training and competency evaluation for medical trainees and practitioners.

Despite the growing need to train medical professionals in virtual care, of the most recent available data from 2021–22 shows that less than sixty percent of medical schools and nursing schools included any formal training in virtual healthcare delivery [15, 16], and even fewer physician assistant (PA) programs incorporated formal telehealth curricula [17]. Unfortunately, those programs that include telehealth training implement their courses as electives and without robust evaluation of educational outcomes [18]. This lack of undergraduate preparation for telehealth care means that most telehealth practitioners have never been formally taught how to best care for their patients remotely [19, 20], creating a training gap that should be addressed urgently.

Simulation-based training has proven to be an effective educational method in healthcare, as clinicians can practice and refine their communication skills in a risk-free environment, and standardized patient (SP) actors have been increasingly utilized to conduct realistic simulated telehealth encounters for training and assessment [21–28]. The use of standardized patients to deliver feedback as a primary component of education has been extensively studied and is known to be an effective technique for adult learning in simulated environments, including in telehealth simulation [29–32]. However, most descriptions of telehealth training in the literature that utilize SP actors involve undergraduate trainees—depictions of SP-based training for the practicing clinician are growing but are currently scant [6, 29, 33–36], and there is a shortage of specific, evidence-based information in the literature about communication in telehealth care for practicing clinicians [37, 38]. Our team's previous experience with video-based simulated telehealth encounters indicated that practicing clinicians perceive these types of programs as enhancing their confidence and skills in using telehealth [29]. We focused on communication skills development in this study as we recognized the significant risk of perpetuating inequity in healthcare, and telehealth may add an additional layer of complexity to these interactions that prior formal communication training may not have adequately addressed [39–43]. Furthermore, to date no studies have focused on training programs specifically oriented to incorporate issues of equity and access in the telehealth—a key domain within the AAMC telehealth competencies that is important yet currently underexplored [8]. The incorporation of telehealth in clinical practice has the potential to exacerbate inequity; our team believes that formal, intentional educational efforts to combat this are needed [39, 41].

This pilot study sought to address the telehealth communications training gap by creating a simulation experience for practicing clinicians, suitable for both physicians and advanced practice practitioners (APPs). This training session was intentionally designed as a standalone, time-efficient simulation that can be implemented for busy clinicians, allowing them to

acquire and practice equity-focused telehealth skills with minimal time investment. We aimed to assess the efficacy of an SP-delivered educational experience in which the SP provided both the portrayal of the patient and the generation and delivery of feedback to the practicing clinicians as an educational intervention. Our primary outcome was the practicing clinicians' checklist-based performance on AAMC-aligned telehealth competencies, specifically centered on Domains II and III (equity, access, and communication) [14]. Secondarily, we captured the participants' perceptions of knowledge acquisition via pre- and post-session survey.

## Methods

### Study design

This study was a prospective interventional study that received pilot funding from the AAMC Telehealth Equity Catalyst (TEC) grant. Utilizing Kolb's Experiential Learning conceptual framework [44–46], an SP-led one-hour simulation-based learning experience was designed consisting of two cases. Learning objectives (S1 Appendix) for the sessions were designed to map to the AAMC telehealth competencies [14]. Each participant completed both cases (A and B), in a cross-over design with half doing Case A first and half doing Case B first. Standardized patient feedback was given to participants after each case by the SP. Participants were surveyed before and after the experience, and their performance was recorded in performance checklists.

### Setting and participants

Eligible participants were physicians or APPs within a fourteen-hospital medical system serving a large urban area in the Northeastern United States practicing in generalist specialties, including emergency medicine, virtual urgent care, internal medicine, and primary care. We did not include pediatrics in this study for consistency of the simulated telehealth patient encounters. Subjects participated on a voluntary basis and were scheduled at times convenient to their schedules. There were no exclusion criteria. Study participants were recruited via email between January and April 2023. The recruitment email outlined the study, encompassing both the potential risks and benefits. Additionally, it reminded participants of their right to withdraw from the study at any point. Data was collected contemporaneously. We trained approximately 8 standardized patients in a recorded session. These SPs were selected by the contracted standardized patient company primarily based on their availability and prior experience in telehealth simulation. Any SP who was not present for the in-person training session was able to view the session recording asynchronously to ensure consistency in their portrayal and in their grading of participants based on the checklist. We aimed for the recruitment of 30 individuals for participation to comply with budgetary limitations in this study. Participants' consent was implied by participation in the study, and the requirement for formal written documentation of consent was waived by the institutional review board (IRB). This study was approved by the IRB at Massachusetts General Hospital (Agreement number 2022A006072).

### Case design

We designed two cases (S2 and S3 Appendices) suitable to a generalist / urgent care environment, intentionally developed by our study team to surface issues of communication and equity in healthcare access. Case A was a case of a 56-year-old person with hypertension seeking care for headaches, but with additional life stressors including juggling the potential loss of a job, caring for her grandchildren, and participating in an e-learning college program. Case B was a 56-year-old person with a history of asthma who had been missing work due to frequent

exacerbations, but with difficulty affording medication refills. The cases were written and iterated from scratch by experts in telehealth, standardized patient case design, ethics, and diversity, equity, and inclusion. The cases were not based on real-life patient encounters.

Prior to the first live session with participants, we held a training session with the SP actors to ensure consistency in the actor portrayal. During this session, the cases were also pilot tested by the SPs, who provided feedback on clarity, flow, and feasibility. As the SPs were expected to complete checklists and provide feedback after each simulated encounter, this activity was modeled and practiced by the SPs. We reviewed the cases with the SPs for clarity and adjusted their timing to fit in the ten minutes allotted for each simulation.

## Data sources and instrument design

Prior to the simulation sessions, participants received learning objectives and a pre-session survey (S4 Appendix). Surveys were administered using REDCap (Vanderbilt University, Nashville, TN). There were no other pre-session requirements for the session, including no pre-session didactics, minimizing the time requirements for the clinicians who volunteered to participate in this study outside of regular work hours. Sessions took place via Microsoft Teams (Microsoft Corporation, Redmond, WA).

Each session included two participants and two trained SPs. Each encounter lasted approximately 10 minutes, immediately followed by 7 minutes during which the SP delivered immediate feedback to the participant. We created a learning objective-aligned checklist to assess performance and communication skills of the participants, which was modified from the Kalamazoo Essential Elements Communications Checklists [47, 48]. This checklist was chosen as a template for our project because previous studies have demonstrated evidence of validity, including in modified forms [49–51]. The checklist contained a total of 22 assessment items (S5 Appendix). The checklist was iteratively refined by the study team through small group sessions and asynchronous reviews, and further adjustments were made based on feedback from the SPs to enhance clarity and usability. Participants' performance was assessed in each of their two cases.

The SP completed the checklist in REDCap (Vanderbilt University, Nashville, TN) simultaneous to or immediately after providing feedback at the end of each case. This allowed feedback to be aligned with learning objectives and, therefore, the AAMC competencies [14]. Participants then performed their second 10-minute case, followed again by feedback. After the session, participants completed a post-session survey (S6 Appendix). The surveys were designed to gauge participants' self-perception of learning objective-aligned skills as a measure of growth, as well as demographic data. A crossover design was utilized to reduce the risk that measured improvement could be due to unmeasured differences in difficulty between the two cases.

## Data analysis

Data was analyzed using Stata (StataCorp, College Station, Texas, version 17) and Microsoft Excel (Microsoft Corporation, Redmond, WA). Descriptive statistics were generated to understand the demographics of the study participants. Paired T-tests were used to test for differences in pre- and post-session survey responses and to evaluate differences in participants' checklist performance between cases one and two.

## Results

A total of 30 clinicians participated in the study (Table 1). Participants included six physicians (20%) and 24 advanced practice providers (NPs or PAs). Participants ranged in experience (0–

**Table 1. Participant characteristics.**

| | |
|---|---|
| **Gender** (n = 30) | |
| Male | 9 (30%) |
| Female | 21 (70%) |
| Non-Binary, Other | 0 (0%) |
| **Physician or Advanced Practice Provider** (n = 30) | |
| Physician | 6 (20%) |
| Advanced Practice Provider | 24 (80%) |
| **Physicians: How many years have you been in practice since residency?** (n = 6) | |
| 0–5 years | 2 (33.3%) |
| 5–10 years | 1 (16.7%) |
| 11–15 years | 1 (16.7%) |
| 16–20 years | 0 (0%) |
| 20+ years | 2 (33.3%) |
| **Advanced Practice Providers: How many years have you been in practice?** (n = 24) | |
| 0–5 years | 11 (45.8%) |
| 5–10 years | 10 (41.7%) |
| 11–15 years | 2 (8.3%) |
| 16–20 years | 1 (4.2%) |
| 20+ years | 0 (0%) |
| **Have you provided patient care via video-based telehealth in the past?** (n = 30) | |
| Yes | 15 (50.0%) |
| No | 14 (46.7%) |
| (blank) | 1 (3.3%) |
| **With which programs have you provided telehealth?** (n = 30) | |
| Partners Healthcare on Demand | 4 (13.3%) |
| Virtual Observation Unit | 1 (3.3%) |
| iPads in the ED during COVID | 1 (3.3%) |
| Virtual Visits for Primary Care Patients | 6 (20.0%) |
| MGB Virtual Urgent Care | 9 (30.0%) |
| Hospital-at-Home | 0 (0.0%) |
| AFC Urgent Care | 1 (3.3%) |

20+ years out of training), and approximately half had participated in telehealth in the past (15, one missing response). All participants (100%) completed all pre and post surveys, and all performance checklists (100%) were completed by the SPs.

## Primary objectives

Our primary objective was performance changes between cases as measured by the checklist. Overall, baseline performance for most checklist items for their first case was quite high, and in many cases all 30 participants met the correctly performed checklist items in both of their cases (Table 2). Some item ratings demonstrated statistically significant improvements after receiving SP feedback on the first case. Specifically, improvement in performance was seen for "ensures my privacy by making sure that my space is private for me" ($p = 0.0226$), "ensures my privacy by making sure and indicating they are in a private space for their conversation (e.g., nobody else can hear our conversation on their end)" ($p < 0.01$), and "ensures that I have access to resources that will support my post-encounter care" ($p = 0.0117$).

**Table 2. Checklist performance by case.**

| Skill/Item Description | Case 1: (% Learners Who Completed Skill [SE]) | Case 2: (% Learners Who Completed Skill [SE]) | Number of Paired Observations | t-value (df) | p-value |
|---|---|---|---|---|---|
| 1. Ensures my privacy by making sure that my space is private for me | 26.7 (8.2) | 60.0 (9.1) | 30 | 2.41 (29) | **0.0226*** |
| 2. Ensures my privacy by making sure and indicating they are in a private space for their conversation with me (e.g., nobody else can hear our conversation on their end) | 23.3 (7.9) | 63.3 (8.9) | 30 | 3.03 (29) | **0.0052*** |
| 3. Avoids background noise | 100 | 100 | 30 | N/A | N/A |
| 4. Uses appropriate lighting so that I can see them | 100 | 100 | 30 | N/A | N/A |
| 5. Turns off other applications (e.g., no other notification noises from emails or messages) | 100 | 100 | 28 | N/A | N/A |
| 6. Adjusts camera to eye level so that I can see their face | 100 | 96.6 (3.4) | 29 | -1.00 (28) | 0.3259 |
| 7. Dresses professionally | 89.7 (5.8) | 96.6 (3.4) | 29 | 1.00 (28) | 0.3259 |
| 8. Begins information exchange by creating relaxed, empathetic environment that promotes good exchange between myself (the patient) and clinician | 93.3 (4.6) | 100 | 30 | 1.44 (29) | 0.1608 |
| 9. Uses non-judgmental statements when communicating with me | 100 | 100 | 30 | N/A | N/A |
| 10. Narrates and explains their actions (e.g., if they need to look at another screen while on the visit) | 100 | 92.9 (7.1) | 14 | -1.00 (13) | 0.3356 |
| 11. Speaks clearly and deliberately so that I can understand | 100 | 100 | 30 | N/A | N/A |
| 12. Uses non-verbal language to show they are listening to me | 100 | 100 | 29 | N/A | N/A |
| 13. Uses pauses to facilitate bilateral communication (listening to me, observing me), allowing patient to contribute to information exchange | 100 | 100 | 29 | N/A | N/A |
| 14. Suggests escalation of care (e.g., go to the ED, visit in-person) if clinician believes I am unsafe (or I express that I feel unsafe) with distance care plan or in my current environment | 100 | 100 | 6 | N/A | N/A |
| 15. Ensures that my care is concordant with my preferences and values | 100 | 100 | 30 | N/A | N/A |
| 16. Explores whether I have social supports and incorporates them as able (if in line with my wishes) | 48.3 (9.4) | 69.0 (8.7) | 29 | 1.44 (28) | 0.1609 |
| 17. Thoroughly and accurately educates me about my illness, its management, and possible consequences with sensitivity to my concerns and preferences | 96.7 (3.3) | 100 | 30 | 1.00 (29) | 0.3256 |
| 18. Verbalizes and clarifies post-encounter plans for my care | 96.7 (3.3) | 100 | 30 | 1.00 (29) | 0.3256 |
| 19. Ensures that I have access to resources that will support my post-encounter care | 80 (7.4) | 100 | 30 | 2.69 (29) | **0.0117*** |
| 20. Adjusts physical examination to the virtual environment | 85.7 (14.3) | 100 | 7 | 1.00 (6) | 0.3559 |
| 21. Guides the patient through physical exam maneuvers | 72.7 (14.1) | 81.8 (12.2) | 11 | 0.56 (10) | 0.5884 |
| 22. Collects/uses the data captured by the patient (e.g., vital signs such as heart rate or where the patient reports pain) | 71.4 (8.8) | 81.0 (10.1) | 21 | 0.62 (20) | 0.5402 |

- SE: Standard Error
- Asterisk denotes statistical significance.
- "N/A" indicates no statistical test was conducted because there was no change in performance between cases 1 and 2.
- Statistical analysis was only conducted in cases where a participant had a value other than "N/A" recorded for both case 1 and case 2 for a particular item. If there was not a recorded value for both cases, that individual was dropped from the analysis. This is why some items have a full 30 values, while others have as few as 6.

## Secondary objectives

Overall, participants reported that they would recommend this training experience, with a mean of 8.8 (SD 1.49) on a Likert scale from 0 (Not at all likely) to 10 (Extremely likely). Participants' responses to every survey question demonstrated a statistically significant improvement

**Table 3. Participants' self-perceived performance pre- and post-session.**

| Attitudes | | | | | |
|---|---|---|---|---|---|
| Question | Pre-Training Mean (SE) | Post-Training Mean (SEs) | Number of Responses | t-value (df) | p-value |
| I can describe at least two ways to adjust physical characteristics (physical space, camera, lighting, microphone) to ensure that the patient experiences a safe environment for a video-based telehealth encounter. | 3.87 (0.19) | 4.67 (0.15) | 30 | -3.38 (29) | **0.0021*** |
| I can describe at least two ways that I can use words/language/dialogue to ensure that the patient experiences a safe environment for a video-based telehealth encounter. | 3.97 (0.11) | 4.53 (0.17) | 30 | -2.81 (29) | **0.0088*** |
| I can give two examples of techniques to create a therapeutic rapport via telehealth by using verbal communication techniques and nonverbal behaviors. | 3.93 (0.13) | 4.63 (0.15) | 30 | -3.43 (29) | **0.0018*** |
| I feel confident applying language that partners with the patient to ascertain and mitigate any risks or unsafe conditions related to the patient's care. | 3.83 (0.17) | 4.50 (0.18) | 30 | -2.66 (29) | **0.0126*** |
| I can describe at least two ways to inquire about and include a patient's family/social support to enhance care during and after a telehealth encounter. | 3.87 (0.16) | 4.43 (0.17) | 30 | -2.43 (29) | **0.0216*** |
| I can describe language techniques to ensure mutually understood post-encounter care plans with my patient and the accessibility of care needs before concluding a telehealth encounter. | 3.90 (0.18) | 4.57 (0.18) | 30 | -2.82 (29) | **0.0086*** |

- SE: Standard Error
- Asterisk denotes statistical significance.

in self-perceived performance/skill level in all learning-objective aligned items (Table 3). The responses to the question, "How likely are you to recommend this simulated telehealth experience to your colleagues?" on a scale of 1 to 10, showed a minimum score of 5, a maximum score of 10, a median score of 9, a mean score of 8.8, and a standard deviation of 1.49 (n = 30).

## Discussion

As telehealth education continues to assume a more central role in health professions education, studies that demonstrate evidence for effective training are timely and needed. In this pilot study, we conducted a training program for 30 practicing clinicians. We found that the use of SP-generated feedback as a primary educational strategy resulted in improved performance as measured on learning-objective aligned checklists, with improvement in multiple domains related to equity and access in healthcare delivery reaching statistical significance. The participants rated the learning experience highly and, when surveyed, endorsed an improvement in confidence and skills for all measured learning objectives. We believe this to be the first study to demonstrate improved telehealth communication performance via telehealth simulated encounters for patient equity and access in healthcare delivery. This is an important area for future research that should be replicated and expanded upon to raise telehealth as a model for providing healthcare to patient populations who may otherwise have limited access. It is possible that this finding could be generalized from practicing clinicians to those in training.

One key aspect we sought to assess in our study was whether a simulation session without pre-session didactics could be effective. As a pilot, it was deemed beyond the scope of the study to develop a series of didactics, but it was also of interest to our team to assess if a shorter training approach could engender measurable learning and change for the learners. Traditionally, pre-session didactics have been included as a part of a module or course [52–54]; however, these studies typically are intended for medical trainees and not practicing clinicians [21, 22, 24, 26, 52–54]. A key innovation of this study is the development of a simulation-based

training that is standalone, requiring minimal time investment from clinicians. This makes it particularly well-suited for busy healthcare professionals, allowing for the integration of equity and access training without the need for extensive pre-session didactics or lengthy time commitments. Typically courses with pre-session didactics require more time and may be less feasible for practicing clinicians who have limited training time. Our study suggests that, at least in some circumstances or for some competencies such as those focused on equity and access, simulation sessions that include focused SP feedback can stimulate behavior change with less time burden. Future research could continue to investigate short simulation sessions using a variety of case complexities, such as handling sensitive social determinants of health or managing challenging patient interactions, to further explore the effectiveness of one-hour simulation sessions for training busy practicing clinicians.

This study demonstrates the feasibility of an actor-run session for telehealth communication training. Any software that allows for video calls, such as Microsoft Teams (Microsoft Corporation, Redmond, WA) or Zoom (Zoom Video Communications, San Jose, CA) could be utilized for sessions at low or no cost. The primary resource for carrying out this type of session is the availability of funds to pay SPs, a cost that should be similar to other simulation sessions that utilize SP services. In our study, the trained SPs proved able to deliver feedback that improved performance on the participants' subsequent case. Thus, it is possible that trained SPs may be capable of teaching this skillset with reduced faculty or instructor involvement, enhancing the efficiency of resource use. The use of SP feedback as a modality for instruction has been well described in the literature [29, 30–32, 55], but this is the first study to our knowledge that demonstrates improved telehealth performance via this approach. In addition, it appears to be the first to focus on training in patient equity and access topics in telehealth. Future research could build upon these findings by incorporating recently-developed competency instruments [56] to validate the results and explore the conditions under which SPs, independent of direct input of medical educators or simulation leaders, may be able to teach telehealth communication skills.

Given the scarcity of educational literature evaluating teaching methods for telehealth communication for practicing clinicians, innovative and evidence-based techniques are urgently necessary. Our study shows that communication skills for patient equity and access in healthcare can be taught efficiently to health professionals. This method of training may be generalizable to other telehealth competencies, such as technology failures or legal / ethical issues in telehealth and it serve as a mechanism for training in other allied health professions. Future research is warranted to explore its efficacy across diverse healthcare settings, evaluate its adaptability for various professional roles, and determine its impact on improving telehealth delivery and patient outcomes.

## Limitations

As a pilot study, the sample size was small and used no control group, limiting generalizability to other settings. Since participation in this study was voluntary, clinicians who were particularly interested or who felt particularly inexperienced in telehealth may have volunteered, contributing to selection bias, which may have increased the measured effect size of the intervention. Although we sought representation from multiple generalist specialties, these results may not be generalizable to all specialties or non-academic urban medical centers. Limited resources prevented a measurement of the durability of learning, an area to direct future study. Additionally, the study did not include a formal needs assessment, which could have provided further insights into the specific educational needs of practicing clinicians in telehealth communication. Finally, although we feel that the use of a modified Kalamazoo

Essential Elements Communications Checklists was logical and similar approaches have been taken in the past [24], any modification of an instrument threatens its validity.

## Conclusion

This pilot study underscores the potential of simulation-based telehealth training for practicing clinicians. The findings suggest that such training can effectively enhance telehealth communication skills and address issues of equity and access in virtual healthcare delivery. The study also demonstrates participant satisfaction with actor-run sessions, which could be a cost-effective and time-efficient approach to telehealth training. However, the study's limitations, including its small sample size and lack of a control group, highlight the need for further research. Future studies should aim to validate these findings, explore the durability of learning, and investigate the generalizability of this training approach to other telehealth competencies and settings.

## Supporting information

**S1 Appendix. Learning objectives.**
(DOCX)

**S2 Appendix. Case A.**
(DOCX)

**S3 Appendix. Case B.**
(DOCX)

**S4 Appendix. Pre-session survey.**
(DOCX)

**S5 Appendix. Standardized patient checklist.**
(DOCX)

**S6 Appendix. Post-session survey.**
(DOCX)

**S1 Dataset.**
(XLS)

## Acknowledgments

The authors thank the Association of American Medical Colleges for their help and guidance in conducting this research.

## Author Contributions

**Conceptualization:** Christopher J. Nash, Susan E. Farrell, Jossie A. Carreras Tartak, Alexei Wagner, Lea C. Brandt, Emily M. Hayden.

**Data curation:** Christopher J. Nash.

**Formal analysis:** Christopher J. Nash.

**Funding acquisition:** Christopher J. Nash.

**Investigation:** Christopher J. Nash, Susan E. Farrell.

**Methodology:** Christopher J. Nash, Susan E. Farrell, Jossie A. Carreras Tartak, Lea C. Brandt, Emily M. Hayden.

**Resources:** Christopher J. Nash.

**Software:** Christopher J. Nash.

**Supervision:** Emily M. Hayden.

**Visualization:** Christopher J. Nash.

**Writing – original draft:** Christopher J. Nash.

**Writing – review & editing:** Christopher J. Nash, Susan E. Farrell, Jossie A. Carreras Tartak, Alexei Wagner, Lea C. Brandt, Emily M. Hayden.

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
