## [Decision Letter · Decision Letter 0]

21 Oct 2024

PONE-D-24-14731Telehealth Equity and Access Communication Skills Pilot Simulation for Practicing CliniciansPLOS ONE

Dear Dr. Nash,

Thank you for submitting your manuscript to PLOS ONE. After careful consideration, we feel that it has merit but does not fully meet PLOS ONE’s publication criteria as it currently stands. Therefore, we invite you to submit a revised version of the manuscript that addresses the points raised during the review process.

We look forward to receiving your revised manuscript.

Kind regards,

Rawshan Jabeen, MHPM, MSc.

Academic Editor

PLOS ONE

**Journal Requirements:**

This research was financially supported by the Association of American Medical Colleges Telehealth Equity Catalyst Grant (https://www.aamc.org/news/telehealth-equity-catalyst-awards), grant number N/A, with authors CJN SEF and EMH as recipients. The funder had no role in study design, data collection and analysis, decision to publish, or preparation of the manuscript.

**Additional Editor Comments:**

Based on my assesment: this article need major revisions.

1. The manuscript needs thorough language refinement for clarity, coherence, and readability. Consider professional editing services.

2. Add more recent references to better support the research gap. Ensure relevant studies, especially from the last five years, are included.

3. Improve data presentation with clearer visuals like tables or figures. Enhance the interpretation by aligning results with the research objectives and ensure the discussion is distinct from the results section.

Decision : The article holds potential, but the suggested revisions are essential for improving the quality and impact of the study. I recommend addressing the issues outlined above before resubmitting for further consideratio

Reviewers' comments:

Reviewer's Responses to Questions

**Comments to the Author**

1. Is the manuscript technically sound, and do the data support the conclusions?

Reviewer #1: Partly

2. Has the statistical analysis been performed appropriately and rigorously? 

Reviewer #1: I Don't Know

3. Have the authors made all data underlying the findings in their manuscript fully available?

Reviewer #1: No

4. Is the manuscript presented in an intelligible fashion and written in standard English?

Reviewer #1: Yes

5. Review Comments to the Author

**Reviewer #1:** Thank you very much for the opportunity to peer-review your article “Telehealth Equity and Access Communication Skills Pilot Simulation for Practicing Clinicians”. It is very well written and aim of this study quite relevant to the field. .

However, I do have some concerns regarding the reporting of the intervention being tested and display of the results and recommend therefore major revision regarding the following comments, divided into the sections of the article.

I look forward to your revision. Thank you very much.

Style:

There are some minor sentences/words where I suggest different grammar use or sentence style:

- Page 4, lll 63-65: “Our team’s previous experience with video-based simulated 64 telehealth encounters demonstrated that practicing clinicians believe that this type of program 65 builds confidence and skills in the use of the telehealth modality [22].” Suggestion: “Our team’s previous experience with video-based simulated telehealth encounters indicated that practicing clinicians perceive these type of program as enhancing their confidence and skulls in using telehealth modality [22].”

- Page 4, ll 65-69: “We focused on 66 communication skills for this study because we recognized that there is significant risk of perpetuating inequity in healthcare, and telehealth may add an additional layer of complexity for these interactions that prior formal communications training may not have adequately addressed.” Suggestion: “We focused on communication skills development in this study as we recognized the significant risk of perpetuating inequity in healthcare, and telehealth may add an additional layer of complexity to these interactions that prior formal communications trainings may not have adequately addressed [30–34].

- Page 6, l 115: “intentionally crafted by our …”, Suggestions: “intentionally developed …”

- Page 6, line 118: “… and participating in online school.”. Who was participating in an online school? The grandmother? For me as a non US speaker, this would be bit unusual as “grandmothers do not go to school anymore. Maybe using “participating in an e-learning college programme.” If the grandkids participate in the online school, I would describe it as following. “ … caring for their grandchildren, who participate in an online school programme.”

Methods:

- General thought: did you pilot test the cases prior to the study? If not, please state. Also, were patients involved in the development?

- How many SPs did you plan on training and to be part of the study? Please describe how you selected these and which criteria were applied

- Instrument Design: How did you develop the SP survey? Can you please provide us with a bit more information here, e.g. how many items, what is the foundation of the development? Also, did you test the instruments before using it in this study?

- Curriculum development and teaching: I am missing the section of how the curriculum looked like and how it was thought? This in my opinion is the most important part of this pilot study, as it is the new intervention being developed and this is completely missing besides Appendix A with the objectives. Please provide an additional section on this in your methods section. Thank you.

Results:

- The way you present the results in table 2 with the SE values as well as p-values is a bit unusual and for me confusing. Could you please not provide results in the table in a more standardized statistically way, e.g. not repeating SE within the cells/results, having an additional p-value column or marking it by * or ** and giving annotations below the table?

- I also have trouble understanding what the wording “Percentage who performed” represents. Does this mean, 26,7% showed the skill “Ensures my privacy by making sure that my space is private for me”? And then the way the p-value is tucked under case 2. Please update the table in a more statistically sound way, e.g.,, with the question if the n differed between pre- and post-test? If not, than you can delete the post-test column and info behind n

- Also, I miss for the t-test the t-value, df…could you please report these?

Table 2

Skill/Item Pre-test (%, [SD]) Post-test (%, [SD]) n (Pre-test) N (Post-test) t-value (df) p-value

Skill 1 26.7 [68.2] 30 28 xxx < 0.05

Annotation: SD – Standard Deviation, df – degrees of freedom

• I do have the same feedback for table 3, please adjust especially the last item of the recommendation into a fourth table, as this way it is really difficult to read and understand.

Discussion:

- Will give peer review once method and results section is clear to me as I do have informations about the intervention yet. Thank you.

6. PLOS authors have the option to publish the peer review history of their article (what does this mean?). If published, this will include your full peer review and any attached files.

Reviewer #1: No

---

## [Author Response · Author response to Decision Letter 0]

21 Nov 2024

To Whom It May Concern:

I have received the list of recommended changes to the manuscript and have made the requested changes in the manuscript. Details of the changes are included in the file that I uploaded, entitled "Response to Reviewers." 

In the online interface, I am unable to make the requested adjustments to the financial disclosure statement. If the editor is willing to make this change on the back end for us, we would appreciate it. The amended funding statement should be as follows:

“This research was financially supported by the Association of American Medical Colleges Telehealth Equity Catalyst Grant (https://www.aamc.org/news/telehealth equity-catalyst-awards), grant number N/A, with authors CJN SEF and EMH as recipients. The funder had no role in study design, data collection and analysis, decision to publish, or preparation of the manuscript. There was no internal funding received for this study. There was no additional external funding received for this study.” 

Thank you, and I look forward to your response. 

Best,

Christopher J. Nash, MD, EdM

---

## [Editor Report · Decision Letter 1]

4 Dec 2024

Telehealth Equity and Access Communication Skills Pilot Simulation for Practicing Clinicians

PONE-D-24-14731R1

Dear Dr. Christopher J Nash

We’re pleased to inform you that your manuscript has been judged scientifically suitable for publication and will be formally accepted for publication once it meets all outstanding technical requirements.

Kind regards,

Rawshan Jabeen, MHPM, MSc.

Academic Editor

PLOS ONE
---

## [Editor Report · Acceptance letter]

9 Dec 2024

PONE-D-24-14731R1 

PLOS ONE

Dear Dr. Nash, 

I'm pleased to inform you that your manuscript has been deemed suitable for publication in PLOS ONE. Congratulations! Your manuscript is now being handed over to our production team.

Kind regards, 

on behalf of

MS Rawshan Jabeen 

Academic Editor

PLOS ONE
